# Real Time PCR and Culture-Based Virus Isolation Test in Clinically Recovered Patients: Is the Subject Still Infectious for SARS-CoV2?

**DOI:** 10.3390/jcm10020309

**Published:** 2021-01-15

**Authors:** Viviana Manzulli, Giulia Scioscia, Giulio Giganti, Maria Rosaria Capobianchi, Donato Lacedonia, Lorenzo Pace, Dora Cipolletta, Pasquale Tondo, Rosella De Nittis, Valeria Rondinone, Luigina Serrecchia, Antonio Parisi, Domenico Galante, Sergio Lo Caputo, Teresa Antonia Santantonio, Damiana Moschetta, Vitangelo Dattoli, Antonio Fasanella, Maria Pia Foschino Barbaro

**Affiliations:** 1Istituto Zooprofilattico Sperimentale of Puglia and Basilicata, 71121 Foggia, Italy; viviana.manzulli@izspb.it (V.M.); lorenzo.pace@izspb.it (L.P.); dora.cipolletta@izspb.it (D.C.); valeria.rondinone@izspb.it (V.R.); luigina.serrecchia@izspb.it (L.S.); antonio.parisi@izspb.it (A.P.); domenico.galante@izspb.it (D.G.); antonio.fasanella@izspb.it (A.F.); 2Respiratory Diseases Unit, Department of Medical and Surgical Sciences, University Hospital “Policlinico Riuniti”, 71122 Foggia, Italy; giulia.scioscia@unifg.it (G.S.); donato.lacedonia@unifg.it (D.L.); dr.pasqualetondo@gmail.com (P.T.); moschettadam@gmail.com (D.M.); mariapia.foschino@unifg.it (M.P.F.B.); 3Laboratory of Virology, National Institute for Infectious Diseases “L. Spallanzani”, 00149 Roma, Italy; maria.capobianchi@inmi.it; 4Clinical Pathology Department, University Hospital “Policlinico Riuniti”, 71122 Foggia, Italy; rdenittis60@gmail.com; 5Infectious Diseases Unit, Department of Clinical and Experimental Medicine, University Hospital “Policlinico Riuniti”, 71122 Foggia, Italy; sergio.locaputo@unifg.it (S.L.C.); teresa.santantonio@unifg.it (T.A.S.); 6University Hospital “Policlinico Riuniti”, 71122 Foggia, Italy; vitangelodattoli@gmail.com

**Keywords:** cell culture, COVID-19, RT-PCR, SARS-CoV-2, viral transmission

## Abstract

Background. The highly variable manifestation of the COVID-19 disease, from completely asymptomatic to fatal, is both a clinical and a public health issue. The criteria for discharge of hospitalized patients have been based so far on the negative result of Real-Time Reverse Transcriptase Polymerase Chain Reaction (RT-PCR) tests, but the persistence of viral fragments may exceed that of the integral virus by weeks. The aim of our study was to verify the clearance of the virus at viral culture in patients hospitalized for COVID-19 that have clinically recovered but are still positive on nasopharyngeal swab. Methods. The study was conducted in hospitalized patients with positive RT-PCR on nasopharyngeal swab. Patients included were from asymptomatic to severe cases and performed nasopharyngeal control swabbing on day 14 for asymptomatic patient or at least three days after remission of symptoms. RT-PCR positive specimens were sent to a biosafety level 3 laboratory for viral culture. Results. We performed a combined analysis of RT-PCR and a highly sensitive in vitro culture from 84 samples of hospitalized patients. The average age was 46 ± 20.29, and 40.5% of the subjects had radiologically confirmed pneumonia, with average PaO_2_ of 72.35 ± 12.12and P/F ratio of 315 ± 83.15. Ct values for the N gene were lower in the first swab than in the control one (*p* < 0.001). The samples from 83 patients were negative at viral culture, and RT-PCR on the respective supernatants always confirmed the absence of viral growth. Conclusions. Our preliminary results demonstrate that patients clinically recovered for at least three days show the viral clearance at viral culture, and presumably they continued to not be contagious.

## 1. Introduction

Since the outbreak of the severe acute respiratory syndrome coronavirus 2 (SARS-CoV-2) pandemic, the increase in the number of cases that have reported positive to diagnostic tests has increased the impact on healthcare and on global public costs. In several areas, the rise in the absolute number of people tested positive to SARS-CoV-2 has been correlated to a higher number of tests performed in the general population, in accordance with improved contact tracing. As a higher transmission in younger subjects has been observed, as compared to the earlier phases of the pandemic, in the current wave, we are able to directly keep track of a greater number of mild and asymptomatic cases [1].

Studies have shown a wide variability in the sensitivity of Real-Time Reverse Transcriptase Polymerase Chain Reaction (RT-PCR) protocols for the detection of SARS-CoV-2, with a false negative rate ranging from 2% (95% CI, 0.3 to 7.9%) to 39.8% (95% CI, 30.2 to 50.2). This variability seems to be at least partially related to the difference in the reagents (i.e., primer sets, buffers, and enzymes, and laboratory kit components in general) that are used [2], but also to the sample type/location, operator, and sample storage time/temperature, as well as the nucleic acid extraction method used [3].

A crucial point in the management of coronavirus disease 2019 (COVID-19), at patient as well as at population level, is to define the exact kinetics of viral shedding and infectiousness. Little is known about the minimal viral load needed to trigger a state of disease, be it completely asymptomatic or overt COVID-19. Furthermore, we currently do not know the clinical meaning of the detection of a certain level of the virus with current diagnostic molecular tests in a general individual, i.e., whether he or she is a potential carrier and what the odds are that they could be, as well as the probability of them developing an overt syndrome. Conversely, to date, detecting positivity to a test for SARS-CoV-2 in a patient with overt disease does not give clear information about how much he or she is actually contagious. A clear frame could shed new light on the kinetics of the viral replication in each host, and possibly give information about why and how some individuals remain completely asymptomatic while others develop a fatal disease. Most importantly, understanding viral dynamics would have a clear impact on the management of COVID-19 at a population level, as it would form the basis of optimized individual isolation timing and social distancing as well as discharge criteria from the hospital.

In addition, many cases have been described of subjects who remain positive to the RT-PCR on nasopharyngeal swab, even many days after the onset of the disease, despite no longer having symptoms. More interestingly, it is not infrequent that patients who have been discharged with repeatedly negative RT-PCR results show again (weakly) positive RT-PCR results at random control swabs, raising concern about the possible need of again establishing isolation measures to avoid the risk of infection transmission from these subjects who have a “viral clearance” from the respiratory system.

Only a few studies, though, have investigated the correlation between RT-PCR cycle thresholds (Ct) and the actual cultivability of the virus from the same samples in vitro.

The aim of our study is to verify, through viral culture, the true persistence of the active, and therefore potentially infecting, virus during the follow-up of patients affected by COVID-19 when they are clinically recovered, including in the analysis of the correlation between Ct cycles and cultivability of virus in vitro patients showing a broad spectrum of clinical severity, i.e., from asymptomatic to severely ill.

## 2. Methods

This study was conducted in patients hospitalized from 1 August to 30 September 2020 in the Respiratory Diseases and Infectious Diseases Units at the University Hospital “Policlinico Riuniti” in Foggia, Italy. Upon entering the emergency room (ER), chest X-ray, laboratory tests, blood gas analysis, and nasopharyngeal swab for SARS-CoV-2 were performed. Patients were admitted to our ward after a positive RT-PCR result on nasopharyngeal swab (T0).

During the hospital stay they underwent standard care medication (corticosteroids, enoxaparin, mucolytics and antibiotics) and oxygen or non-invasive ventilation, according to their clinical condition.

A nasopharyngeal control swab was performed on day 14 if the patient was completely asymptomatic or at least three days after remission of symptoms (T1) and therefore declared “clinically recovered”. This definition includes complete weaning from oxygen therapy or ventilator support and no main symptoms of infection (no fever, no dyspnea, SaO_2_ stably above 96% in room air), with laboratory values compatible with only iatrogenic effect (e.g., mild leukocytosis from corticosteroids) and mild elevation of D-dimers, though nonetheless highly reduced from the time of admission. In this case, patients could be discharged from hospital if he or she was negative at nasal swab.

This schedule is based on the national discharge guidelines that needed a negative result of RT-PCR performed on nasopharyngeal swabs on two consecutive days. It is also based on a highly improbable negative RT-PCR in the first 14 days of hospital stay. We further investigated four clinically recovered patients who were persistently positive, requiring more than 1 month of hospitalization. In the case of positive RT-PCR, the nasopharyngeal swabs were sent to a biosafety level 3 (BSL-3) laboratory for viral culture.

This study was approved by our Institutional Review Board (Dir 389-20). All patients’ data were collected in the context of routine clinical care, and written informed consent was signed at admission according to hospital policy.

### 2.1. Specimen Processing and Analysis

#### 2.1.1. RT-PCR

Viral RiboNucleic Acid (RNA) was extracted from nasopharyngeal swabs within 2 h of collection, using the STARMag 96 × 4 Universal Cartridge kit with the Microlab NIMBUS IVD instrument, according to the manufacturer’s instructions (Seegene Inc. Seoul, Korea). Amplification and detection of target genes (N, E, and RdRP) was performed using the commercially available kit AllplexTM 2019-nCoV Assay (Seegene Inc. Seoul, Korea) with the CFX96TM instrument (Bio-Rad, Hercules, CA). The cycle threshold (Ct) of each RT-PCR reaction was extracted from the Seegene Viewer software, used for results interpretation, and recorded into a dedicated Excel database. The test was considered positive when at least one of the three investigated genes had a Ct below 40. As the N gene assay resulted 10 times more sensitive that the ORF1b gene for detecting viral infection [4], the viral loads were estimated basing on the Ct values for the N target using the ΔCt method (Ct sample—Ctref: = ΔCt − N), as previously described [5].

#### 2.1.2. Virus Isolation

For SARS-CoV-2 isolation, the Vero E6 cell line (African green monkey kidney cells) was used [6,7]. Cells were cultured in Eagle’s minimal essential medium (EMEM) (Life Technologies, Carisbad, CA, USA) supplemented with 10% (vol/vol) fetal bovine serum (FBS) (Life Technologies, Carisbad, CA, USA), and 100 U/mL penicillin and streptomycin (Life Technologies, Carisbad, CA, USA).

For the virus isolation from swab, cells were plated into a 25 cm^2^ cell culture flasks (Corning, CLS430168) at a confluency of 70–80% in 6 mL EMEM with 6% FBS and incubated overnight at 37 °C. The following day, 1500 μL of the swab medium was incubated with 500 μL of an antibiotic solution (2000 U/mL of penicillin/streptomycin and 300 U/mL of neomycin) for 1 h at room temperature. The 2 mL of suspension was then inoculated on the monolayer of the VeroE6 cells. The flask was incubated at 37 °C for 1 h.

After incubation, 4 mL of EMEM with 6% fetal bovine serum (FBS) was added and incubated again at 37 °C for 72 h [8]. Each day, 200 μL of EMEM were collected from each flask for biomolecular testing. The EMEM 6%FBS was replaced every 72 h. The observation lasted for a week. The result was defined on the basis of cytopathic effect (subjective reading) combined with the positivity of the RT-PCR test (objective reading) in supernatant. All procedures for viral culture followed the laboratory biosafety guidelines.

#### 2.1.3. Definition of the Sensitivity Level of the Isolation Test

To establish a sensitive procedure for virus isolation in cell culture, two viral suspensions obtained from two SARS-CoV-2 Italian patients containing known virus concentrations (100 TCID_50_) were tested.

Serial dilutions of both suspensions were made to obtain infectious viral particles at various concentrations in 2 mL of EMEM containing 6% FBS. Each virus dilution was seeded on a Vero E6 cell monolayer and incubated at 37 °C. After one hour of incubation, a further 4 mL of EMEM 6% FBS was added and, at the same time, 200 μL of the supernatant were taken, on which the RNA extraction and the RT-PCR test were carried out to verify the value of the starting Ct. For each viral dilution, four tests were performed.

Every 24 h, the infected cell monolayer was visually inspected by light microscopy to check for the presence of cytopathic effect, and 200 μL of supernatant were collected, on which the RNA extraction and the RT-PCR test were carried out to verify the Ct value. After 72 h from the beginning of the incubation, the medium was replaced.

For each time point, one step reverse transcription PCR was performed to detect the RdRp gene, as described by Corman, V.M. et al. [9], as this target is the most specific one for the detection of Sars-CoV-2. Briefly, viral RNA was extracted from medium by QIAamp Viral RNA Mini Kit (Qiagen, Hilden, Germany), following the manufacturer protocol.

A 25-μL reaction was set up containing 5 μL of RNA, 12.5 μL of the 2× reaction buffer provided with the Superscript III one step RT-PCR system with Platinum Taq Polymerase (Invitrogen, Carlsbad, CA, USA), 1 μL of reverse transcriptase/Taq mixture from the kit, 0.4 μL of a 50 mM magnesium sulfate solution, 1.5 μL of Forward primer (10 μM, GTGARATGGTCATGTGTGGCGG), 2 μL of Reverse primer (10μM, CARATGTTAAASACACTATTAGCATA), and 0.5 of probe (10 μM FAM-CAGGTGGAACCTCATCAGGAGATGCBBQ).

All oligonucleotides were synthesized and provided by Invitrogen (Carlsbad, CA, USA). Thermal cycling was performed at 55 °C for 10 min for reverse transcription, followed by 95 °C for 3 min and then 45 cycles of 95 °C for 15 s and 58 °C for 30 s.

Overall, the experiment lasted 120 h. Infectious virus was considered to be present in those wells that showed a combined reduction of the Ct value and the presence of a cytopathic effect. Isolation in flasks with 10, 5, and 1 viral infecting doses occurred within 72 h in all cases. In the tests carried out using 0.5 and 0.1 viral infecting doses, only one isolation occurred within 72 h in the 0.5 dose group, and one isolation occurred at 96 h in the 0.1 dose group (Table 1 and Table 2).

### 2.2. Statistical Analysis

All analysis was performed using SPSS 26 software (IBM, Armonk, NY, USA). Numerical data were represented with mean ± standard deviation, while categorical variables were represented as counts and percentage. Any statistical differences in the laboratory data of population at baseline and nasopharyngeal control swab were performed using *t*-test paired samples or Mann–Whitney U–test, as appropriate. A *p*-value of less than 0.05 was considered statistically significant.

## 3. Results

We selected 84 subjects whose nasopharyngeal swab was still RT-PCR-positive at T1. Table 3 summarizes the characteristics of the subjects at baseline. The average age was 46 ± 20.29, (19.1% female), 51.2% of patients were African, and the remaining were Caucasian. At admission, 40.5% of the subjects had radiologically confirmed pneumonia, the overall average PaO_2_ was 72.35 ± 12.12, PaCO_2_ 36.2 ± 5.2, with an average P/F ratio of 315 ± 83.15. Five percent of patients underwent oxygen therapy, and 9.6% underwent high flow nasal cannulae (HFNC) or continuous positive airway pressure (CPAP) ventilation.

Nasopharyngeal swabs were performed, on average, on day 19.95 ± 5.71 in 80 patients, while in four persistently positive cases these were performed at day 41, 43, 50, and 70, respectively. As expected, Ct values for the N gene were lower in the first nasopharyngeal swab than in the control one (26.04 ± 5.26 vs. 35.59 ± 3.95, *p* < 0.001—Table 4, Figure 1). The first group of patients became negative after two consecutive swabs, about 10 days after culture (29.50 ± 9.17), while the four persistently positive patients needed 53, 61, 72, and 104 days, respectively, to obtain a double negative response.

Using a sensitive virus culture method able to detect even fractions of viral infectious units, the samples from 83 patients were negative at viral culture, and the RT-PCR on the respective supernatants always confirmed the absence of viral growth (Figure 2a). Only in one case (1.19%) a cytopathic effect was observed in the corresponding cell culture at 96 h post-seeding (Figure 2b), and a relevant reduction of the viral load (increase of Ct from 31 to 38) in RT-PCR was also observed. However, the patient had a presumably delayed clearance because he was affected by Acute Myelogenous Leukaemia (AML) when he acquired the SARS-CoV-2 infection and was under chemotherapy treatment; this subject resulted negative to a nasopharyngeal swab ten days later.

## 4. Discussion

This study shows that the positivity of RT-PCR on nasopharyngeal swab performed during a follow up check in recovered COVID-19 patients about 20 days after symptoms onset does not coincide with the presence of the infectious virus established with the virus culture method. The main implication of these findings is that clinically recovered patients, in >98% of cases, are no longer contagious, despite the presence of vestigial virus components detected by RT-PCR. Since the beginning of the pandemic, many studies have tried to estimate the infectivity according to the clinical features and to the viral load detected by diagnostic RT-PCR [10,11]. According to an analysis made by He et al. [12], the infectiousness of COVID-19 peaks early, around the onset of symptoms, and declines within 8 days, as proven by analyzing serial time intervals between symptoms onset in two individuals in a contagious chain.

The most reliable method for COVID-19 diagnosis is RT-PCR [13] based on the detection of nucleic acid from the SARS-CoV-2 in upper and lower respiratory specimens (such as nasopharyngeal or oropharyngeal swabs, sputum, lower respiratory tract aspirate, bronchoalveolar lavage, and nasopharyngeal wash/aspirate or nasal aspirate). Ct values may be converted to Log10 RNA copies/mL using calibration curves based on quantified sample RNA transcripts, providing a quantitative result [14].

Huang et al. [15] studied 60 specimens from 50 patients, performing RT-PCR for genes E, N, and Nsp12, as well as in vitro virus cultivation on Vero E6 cells. The specimens resulting positive in culture had significantly higher Ct values for all three genes. They further proved that the analysis of structural and non-structural genes in both culturable and non-culturable samples might help to identify actively replicating viruses.

La Scola et al. [16] highlighted a correlation between Ct values and the probability of isolating the virus in vitro; from a group of 183 samples, virus isolation always (100%) occurred in swabs with a Ct between 13 and 17, and viral growth was never observed with Ct > 34.

Arons et al. [17] analyzed specimens from 76 residents in a skilled nursing facility and found that, after classification on symptoms, no significant difference was observed in the Ct values of patients, and there was no significant difference according to symptoms intensity. The analysis was performed with RT-PCR on nucleocapsid gene regions N1 and N2. A parallel in vitro test of viral culture found no virus rescue after 9 days from symptoms onset.

Wölfel et al. [14] performed RT-PCR for E and RdRp genes and in vitro culture of nasopharyngeal swab, sputum, sera, and stool samples from none patients with mild disease. They found that RT-PCR can result positive for as long as 28 days from symptoms onset. No virus could be cultivated from samples collected after 8 days, and could also never be cultivated for viral loads with less than 10^6^ copies of RNA. Despite the small size of the study population, the time for virus isolation success, estimated via a probit model, was 9.78 days (CI 8.45–21.78). Notably, they observed an earlier peak of shedding in the nasal-pharyngeal swab as compared to sputum, and this was higher in intensity in patients with pneumonia. Urine and serum samples never showed viral growth in vitro, while stool samples showed a prolonged viral RNA shedding, extending far beyond the virus isolation window.

Bullard et al. [18] performed a comparison between SARS-CoV-2 viral culture on Vero cells with RT-PCR; they analyzed 90 samples (nasopharyngeal and endotracheal swabs) from patients at different time points from symptoms onset (0 to 21 days) and observed that no virus growth is obtained, either after 8 days from symptoms onset or when Ct > 24 on RT-PCR. For each 1 unit increase of Ct, the odds ratio of infectivity decreased by 32%, while, for each day increase from symptoms onset, the decrease was by 37%. The limits to this study are that only one gene (E) has been analyzed and no longitudinal analysis has been performed.

In a pre-print work by Van Kampen et al. [19], a study of samples from 129 patients with severe/critical disease has shown a possible longer infectious viral shedding as compared to mild and moderate cases. They observed a correlation between viral load and the probability of isolating the virus in vitro. A reduction to less than 5% of this probability was observed for viral loads of less than 10 copies or after 15.2 days after symptoms onset.

In our study, the culture from the swabs of positive patients is able to define the presence of live and viable viral particles within 96 h from the seeding on Vero E6 monolayer cells. Furthermore, the sensitivity of the test is confirmed by the fact that, even with dilutions in which 0.5 and 0.1 infectious viral particles are expected to be present, 25% of the cultures provided successful virus isolation. These results confirm the viral isolation method as the most sensitive and specific one, and it is considered the gold standard according to classical virological methods, capable of discriminating persons that harbor the live infectious virus from those that probably harbor dead virus particles or fragments of viral RNA not yet detected by RT-PCR.

Compared to the studies in the literature, our preliminary results demonstrate that patients clinically recovered for at least three days showed the viral clearance at viral culture, and presumably they continued to not be contagious. There is no doubt that both the RT-PCR and the viral culture have limits, so non-infectivity cannot be excluded with absolute certainty, and genomic and sub-genomic RNAs analysis could provide more evidence about the real infection risk. Deeper studies are necessary to better evaluate this aspect. However, according to our results, and in line with the data that emerged from the other few studies performed on cell cultures, we can suppose that it might not be strictly necessary to perform control swabs to demonstrate the clearance of the virus when patients have clinically recovered for at least three days. Further studies, involving patients with a shorter time of recovery, that is, earlier than 14 days of hospitalization, might conclusively prove this observation and significantly shorten the length of hospitalization as well as time of social isolation.

## Figures and Tables

**Figure 1 jcm-10-00309-f001:**
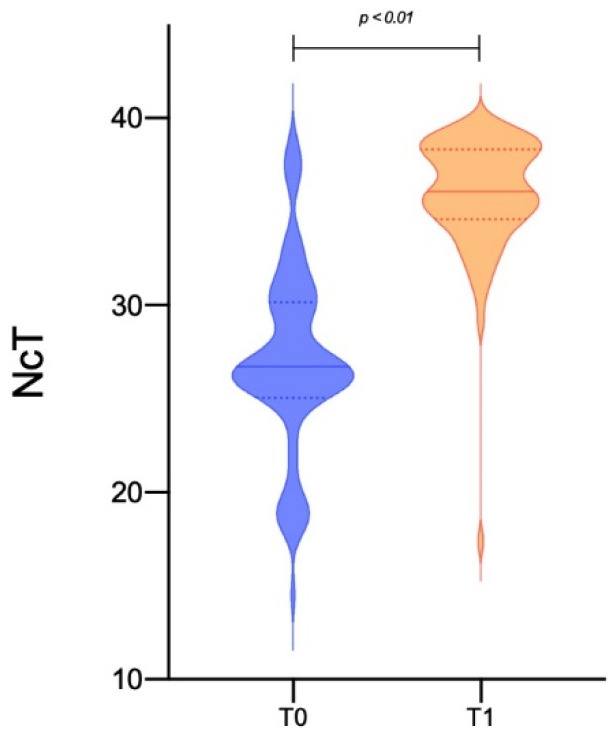
Comparison of cycle threshold (Ct) values at T0 (positive nasopharyngeal swab at hospital admission) and at T1 (nasopharyngeal control swab at clinical recovery performed after 19.95 ± 5.71 days from T0) for the *N* gene in a population of 84 patients described by Violin plot).

**Figure 2 jcm-10-00309-f002:**
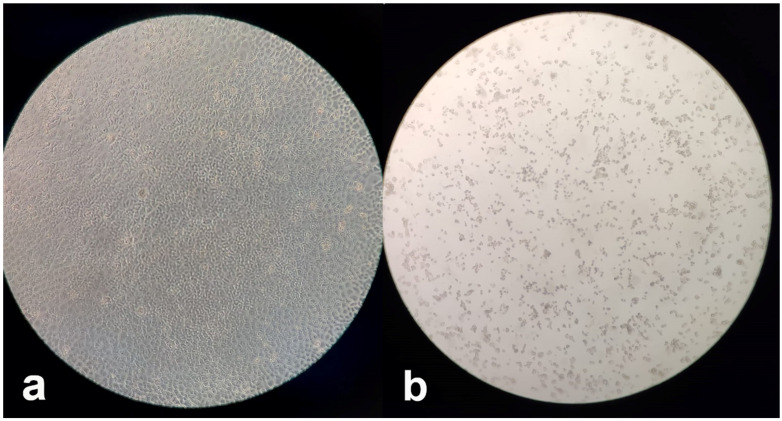
Viral cultures examined by a microscope at 5× magnification: (**a**) Absence of viral growth, as shown by complete confluence of Vero E6 cells; (**b**) Cytopathic effects consisting of rounding and detachment of cells in Vero E6 cell cultures at 96 h post-seeding.

**Table 1 jcm-10-00309-t001:** Data relating to cytopathic effect of each dilution detected at 72, 96, and 120 h of incubation.

Infectious Viral Doses (TCID50)	72 h	96 h	120 h	72 h	96 h	120 h
	Cytopathic Effect	Result of Tests	Cytopathic Effect	Cytopathic Effect	Result of Tests	Cytopathic Effect
10	YES	Positive 4/4—end test				
5	YES	Positive 4/4—end test				
1	YES	Positive 4/4—end test				
0.5	NO	Negative 4/4—continue test	YES NO	Positive 1/4—end test	NO	Negative 3/3—end test
				Negative 3/4—continue test		
0.1	NO	Negative 4/4—continue test	YES NO	Positive 1/4—end test	NO	Negative 3/3—end test
				Negative 3/4—continue test		

Notes: (Median Tissue Culture Infectious Dose (TCID50)).

**Table 2 jcm-10-00309-t002:** Data resulting from the comparison of cytopathic effect and Ct values obtained by PCR analyses in relation to the different infectious viral doses used to standardize the test. The indicated Ct values are the average medium ones, obtained by the four replicates.

Infectious Viral Doses (TCID50)		72 h	96 h	120 h
	Starting Ct (RdRp)	Ct Value (RdRp)	Ct Value (RdRp)	Ct Value (RdRp)
10	27.33	12.31	-	
5	28.60	12.10	-	
1	24.42	17.13	13.73	-
0.5	30.77	32.56	36.02	Negative
0.1	32.67	34.67	34.46	Negative

Notes: Cycle thresholds (Ct) of the *RdRp* gene. Median Tissue Culture Infectious Dose (TCID50).

**Table 3 jcm-10-00309-t003:** Characterization of the study group at baseline (N = 84).

**Patient Characteristics**	
Age, years	46.07 ± 20.29
Females, n (%)	16 (19.1)
African, n (%)	43 (51.2)
Current or former smoker, n (%)	24 (28.6)
**Comorbidities, n (%)**	
Asthma or COPD	3 (3.6)
Chronic cardiac disease	27 (32.2)
Diabetes	10 (12)
Alzheimer	5 (6)
Chronic neurological disease	8 (9.6)
**Radiographic feature, n (%)**	
Pneumonia	34 (40.5)
**Respiratory failure, n (%)**	
Oxygen therapy	13 (15.5)
HFNC or CPAP	8 (9.6)

Continuous data are expressed as mean ± standard deviation, numerical as count (percentage). Notes: COPD: chronic obstructive pulmonary disease; CPAP: continuous positive airway pressure; HNFC: high nasal flow cannulae.

**Table 4 jcm-10-00309-t004:** Comparison between laboratory characteristics of patients at T0 and at T1 (interval of 19.95 ± 5.71 days).

Parameters	T0			T1		
	Min	Max	Mean ± SD	Min	Max	Mean ± SD
Laboratory tests and blood gas analysis						
WBCs (10^3^/uL)	1.068	15.6	6.243 ± 3.209	2.4	13.5	6.013 ± 2.724
Lymphocytes (10^3^/uL)	0.099	2.504	0.993 ± 0.613	1.082	2.541	1.576 ± 0.543 **
D-dimer (ng/mL)	180	5918	1037.68 ± 1206.56	181	1849	691.82 ± 463.45 *
ESR (mm/h)	4	126	40.63 ± 36.543	5	77	30.88 ± 28.62 *
IL-6 (pg/mL)	1.59	139.4	22.55 ± 35.68	1.39	8.2	4.25 ± 2.25 **
CRP (mg/L)	0.1	307.2	57.13 ± 84.64	0.3	108.3	16.41 ± 30.83 **
pH	7.32	7.58	7.44 ± 0.05	7.43	7.51	7.46 ± 0.02 **
PaO_2_ (mmHg)	46	90	72.35 ± 12.12	59	89	76 ± 8.70 *
PaCO_2_ (mmHg)	24	46	36.2 ± 5.21	32	43	36.9 ± 3.84
P/F ratio (mmHg)	110	428.6	315 ± 83	245.8	423.8	358.39 ± 49.73 **
RT-PCR NcT value						
	14.5	38.9	26.04 ± 5.26	17.40	38.9	35.59 ± 3.95 **

Data are expressed as mean ± standard deviation, with the minimum and maximum values found. * *p* < 0.05, ** *p* < 0.01 vs. baseline. Notes: CRP: C-reactive protein; ESR: erythrocyte sedimentation rate; IL-6: Interleukin 6; NcT: cycle thresholds (Ct) of *N* gene; PaCO_2_: partial pressure of arterial carbon dioxide; P/F ratio: arterial partial pressure of oxygen (PaO_2_) to fraction of inspired oxygen (FiO_2_); WBCs: white blood cells.

## Data Availability

The data presented in this study are available on request from the corresponding author.

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
