# Peer review of "Real Time PCR and Culture-Based Virus Isolation Test in Clinically Recovered Patients: Is the Subject Still Infectious for SARS-CoV2?"

_jcm, 2021, doi:10.3390/jcm10020309_

Round 1

Reviewer 1 Report

This is a highly relevant clinical study on the probability of contagiousness in patients with persistently positive PCR results for SARS-CoV-2. The author should be congratulated on their efforts. I have two minor remarks:

1) 84 patients were included in the study out of how many altogether? What were the inclusion criteria - i.e. were ALL pts. with persistently positive PCR included? Any information on wether the patients with repeatedly positive PCR differed from the rest?

2) The one patient with persistent positive viral culture should be discussed in more detail - obviously, in certain settings a Ct value does not predict contagiousness. 

Author Response

Response to Reviewer 1 Comments

This is a highly relevant clinical study on the probability of contagiousness in patients with persistently positive PCR results for SARS-CoV-2. The author should be congratulated on their efforts. I have two minor remarks:

Point 1: 84 patients were included in the study out of how many altogether? What were the inclusion criteria - i.e. were ALL pts. with persistently positive PCR included? Any information on wether the patients with repeatedly positive PCR differed from the rest?
Response 1: Thank you for this observation. The only criteria of admission in the study was the persistent of positive swab even in case of clinical recovered. So, we enrolled all patients who met this criteria during period of observation. Therefore, all of the admitted patients were eligible to this study, and we performed tests for those who completed the entire hospitalisation in our ward. We add this information in the text.
Point 2: The one patient with persistent positive viral culture should be discussed in more detail - obviously, in certain settings a Ct value does not predict contagiousness.
Response 2: Thank you for the observation. We reformulated the part concerning this patient.

Reviewer 2 Report

Thank you for the opportunity to review the manuscript submitted to JCM by Manzulli et al. entitled "Real time PCR and Culture-based Virus isolation test 2 in clinically recovered patients: is the subject still 3 contagious for SARS-CoV2?".

Major comments:

line 43/44 (and 258/259): "Our preliminary results demonstrate that patients clinically recovered from at least 3 days show the viral clearance at viral culture and conseguently they are not still contagious." The conclusion statement cannot be made as there is a limit of detection of both the culture and rt-PCR assays, individuals may be contagious. They also only use one RT-PCR platform and one culture method (VeroE6 cells only) with in-house controls that have not been calibrated to an external reference and the results could only be applied to their laboratory situation and not transferable to other laboratories. In the discussion they note other methods that evaluate the Ct value in relation to the infectious content of the swab give comparable results to theirs.

line 106: "written informed consent was waived", in line with the journals guidelines please ensure there is justification for consent to be waivered. Particularly as a patient may be able to identify themselves from the manuscript information lines 173-175: "he was affected by Acute Myelogenous Leukaemia (AML) and acquired the SARS-CoV-2 infection one month after the 17th cycle of azacytidine" (Azacitidine?)

The authors show the Ct values for each time point (T0 vs T1), this could be represented differently as it is paired data and could be presented as T0-T1 increase Ct, T0-T1 decrease Ct, To-T1 no change in Ct and the Ct of the cultured sample graphically represented.

They also only show the results of one RT-PCR assay whereas in the methods they reference other target genes such as N, E and RdRP using the SeeGene platform. No interpretation of the results according to the manufacturers’ instructions is given, at what Ct is the test considered positive or negative.

There is no discussion as to the role of defective interfering particles that viruses produce that contain nucleic acid but are incapable of infecting cells, this would contribute to the high Ct value with no infectivity in culture.

Minor comments:

Please also check spelling throughout and correct multiple instances of words being concatenated, consistency of spelling and spelling errors, as examples,

line 44: conseguently, please revise to consequently

line 52: contacttracing, please separate words

line 60: kitcomponents, please separate words

line 71: understandingviral, please separate words

line 85: broadspectrum, please separate words

line 94: mucolyticsand, please separate words

line 103: nasopharyngealswabsweresent, please separate words

lines 109/152 vs 156/162: nasofaringeal vs nasopharingeal

line 33/34: end of sentence "The study was conducted in hospitalized RT-PCR nasopharyngeal swab positive. " include patients for clarification.

lines 56-60: Not only are the RT-PCR conditions and components a factor in the test result but also the sample type / location, operator and sample storage time / temperature as well as the nucleic acid extraction method used. Please see other refs such as PMID: 32573469.

line 77: please define "T1"

line 135: Two patient samples were used to test the limit of detection of assays, there are reference materials available that are better characterised than a clinical sample and would enable better standardization and assessment of the limit of detection of the assays and increase the impact of the manuscript.

There are little / no figure legends:

line 179: figure 1 legend requires description, including n values (number in each group).

line 181: figure 2 requires better imaging with scale bars

line 185: for table 1 include more detail as to the number of replicates and occasions this was repeated for establishing quality of this result, it is also important to detail how the clinical samples were quantitated for their "known titre".

Author Response

Response to Reviewer 2 Comments

Thank you for the opportunity to review the manuscript submitted to JCM by Manzulli et al. entitled "Real time PCR and Culture-based Virus isolation test 2 in clinically recovered patients: is the subject still 3 contagious for SARS-CoV2?".

Major comments:
line 43/44 (and 258/259): "Our preliminary results demonstrate that patients clinically recovered from at least 3 days show the viral clearance at viral culture and consequently they are not still contagious."

Point 1: The conclusion statement cannot be made as there is a limit of detection of both the culture and rt-PCR assays, individuals may be contagious. They also only use one RT-PCR platform and one culture method (VeroE6 cells only) with
in-house controls that have not been calibrated to an external reference and the results could only be applied to their laboratory situation and not transferable to other laboratories. In the discussion they note other methods that evaluate
the Ct value in relation to the infectious content of the swab give comparable results to theirs.
Response 1: Thank you for this remark, of course our considerations are the results of in vitro analyses, so we added in text “presumably are not still contagious” as well as we more remark limits of both technics in the discussion. About viral culture we used the most common cell line (VeroE6) for the isolation of SARS-CoV-2, and we used as reference material two SARS-CoV-2 viral suspensions to perform the sensibility of the cell culture test. The first one has been obtained from a patient affected by Covid-19 in Lombardy in the first stage of the pandemic in Italy during this winter. This viral suspension has become a reference material for Italy since it has been titrated (100 TCID50/ml) by San Matteo Polyclinic of Pavia and has been used as standard within an interlaboratory proficiency testing test between italian laboratories. The second one was obtained and titrated by our lab from a 10 days newborn in the second stage of the
pandemic (August) about we already published a paper (DOI:10.1016/j.idcr.2020.e00960). The aim of using two viral suspensions was to compare viruses obtained in two different periods and to confirm the limits of detection of the assay with both of them. On the basis of the above we think that the test can be reproducible for other laboratories.
Point 2: line 106: "written informed consent was waived", in line with the journals guidelines please ensure there is justification for consent to be waivered. Particularly as a patient may be able to identify themselves from the manuscript
information lines 173-175: "he was affected by Acute Myelogenous Leukaemia (AML) and acquired the SARS-CoV-2 infection one month after the 17th cycle of azacytidine" (Azacitidine?)
Response 2: Thank you for this remark, there was a misunderstanding, as we meant that specific consent was not required for the study but because each admission to the hospital requires an informed consent that includes the
authorisation to use the clinical data and samples taken for routine clinical management for research purpose. Which, of course, was taken. In this study with did not perform extra tests nor swabs other than those needed for the
ordinary medical care. We clarify this point in the text (line 116-117). About the possibility that patient who was positive could identify them self we removed some information about him.
Point 3: The authors show the Ct values for each time point (T0 vs T1), this could be represented differently as it is paired data and could be presented as T0-T1 increase Ct, T0-T1 decrease Ct, To-T1 no change in Ct and the Ct of the
cultured sample graphically represented.
Response 3: Thank you for your suggestion. We try to represent data as you suggest but the image in our opinion is not clear (as you can see below) so we prefer let the figure as it is. We did not add the cT of the cultures because it is usually much lower due to the greater amount of RNA and not comparable with the cT of the swabs
Point 4: They also only show the results of one RT-PCR assay whereas in the methods they reference other target genes such as N, E and RdRP using the SeeGene platform. No interpretation of the results according to the manufacturers’
instructions is given, at what Ct is the test considered positive or negative.
Response 4: Thank you for your observation. Seegene platform is more sensitive to N gene, compare to genes E and RdRP, so if the detection of E and RdRp can be useful for diagnosis when patients had a suspicion of infection, for research purpose is better to refer to N ct. For this reason we don’t add the cT value of other two genes which were express less time of N gene.
Point 5: There is no discussion as to the role of defective interfering particles that viruses produce that contain nucleic acid but are incapable of infecting cells, this would contribute to the high Ct value with no infectivity in culture.
Response 5: Thank you for your remark. When there is viral growth associated with cytopathic effect, usually there is a reduction of at least 6-7 Ct in PCR compared to the initial Ct of the swabs. Such a high difference cannot be due to defective interfering particles. Moreover if we don’t have virus growth in the first 72 hours, we totally change medium (EMEM) in the flask, taking away all the RNA residues and we repeat PCR from the day after on a completely new medium. It means that if there are high Ct values in PCR, they are just due to the replication of the virus in the cells. In this way we exclude the defective interfering role of particles that viruses produce.
Point 6: Minor comments:
Please also check spelling throughout and correct multiple instances of words being concatenated, consistency of
spelling and spelling errors, as examples,
line 44: conseguently, please revise to consequently
line 52: contacttracing, please separate words
line 60: kitcomponents, please separate words
line 71: understandingviral, please separate words
line 85: broadspectrum, please separate words
line 94: mucolyticsand, please separate words
line 103: nasopharyngealswabsweresent, please separate words
lines 109/152 vs 156/162: nasofaringeal vs nasopharingeal
line 33/34: end of sentence "The study was conducted in hospitalized RT-PCR nasopharyngeal swab positive. " include
patients for clarification.
Response 6: Thanks for the observations. We have added corrections to the text.
Point 7: lines 56-60: Not only are the RT-PCR conditions and components a factor in the test result but also the sample type / location, operator and sample storage time / temperature as well as the nucleic acid extraction method used. Please see other refs such as PMID: 32573469.
Response 7: Thank you, we add these specifications in the text and the relative reference.
Point 8: line 77: please define "T1"
Response 8: Thank you for the observation. We reformulated the sentence.
Point 9: line 135: Two patient samples were used to test the limit of detection of assays, there are reference materials available that are better characterised than a clinical sample and would enable better standardization and assessment of the limit of detection of the assays and increase the impact of the manuscript.
Response 9: Thanks for the precise observation. The two SARS-CoV-2 viral suspensions used in this study to perform the sensibility of the cell culture test with VERO E6 cells were obtained from clinical samples of italian patients. The first
one has been obtained from a patient affected by Covid-19 in Lombardy in the first stage of the pandemic in Italy during this winter. This viral suspension has become a reference material for Italy since it has been titrated (100 TCID50/ml) by San Matteo Polyclinic of Pavia and has been used as standard within an interlaboratory proficiency testing test between italian laboratories. The second one was obtained and titrated by our lab from a 10 days newborn in the second stage of the pandemic (August) about we already published a paper (DOI:10.1016/j.idcr.2020.e00960). The aim of using two viral suspensions was to compare viruses obtained in two different stages and to confirm the limits of detection of the assay with both of them.
There are little / no figure legends:
Point 10: line 179: figure 1 legend requires description, including n values (number in each group).
Point 11: line 181: figure 2 requires better imaging with scale bars
Response 10/11: Thank you, we reformulated legends for figures 1 and 2, as suggested in your note.
Point 12: line 185: for table 1 include more detail as to the number of replicates and occasions this was repeated for establishing quality of this result, it is also important to detail how the clinical samples were quantitated for their "known
titre".
Response 12: Thank you for your suggestion. The sensitivity level of the in vitro isolation test was repeated four times obtaining the same results. The viral suspensions used were titrated with the limit dilution method and the titer was calculated by Reed and Muench method.

Reviewer 3 Report

Manzulli et al, compare real-time PCR detection of SARS-CoV-2 RNA and culture-based virus isolation to address the important question of viral infectivity during the acute and immediate post-acute infection period. Understanding the kinetics and dynamics of the SARS CoV-2 virus and hence implications for transmission remains an important question. The data presented are derived from cohorts of SARS CoV-2 infected patients in Italy. The authors report on a ‘sensitive’ culture-based assay and compare their read-outs to a quantitative RT-PCR system.

The data generated would appear to support the main finding of the paper although the authors do not present a completely convincing case and there appears to be no clear over-riding novel message driven by a unique data set. As the authors acknowledge other groups have reported  that a PCR Ct  value >24, in a patient >d8 post infection is likely to be non-infectious. As the outcome  of any such study carries a significant public health message, complete rigour in scientific approach and execution must be applied.

The methodology describing the culture side appeared to lack certain key details making it difficult for the reader to fully appraise the method used. The authors need to be more precise in their descriptions of methodology throughout.

‘Clinically recovered’ is a fairly imprecise definition when dealing with potential asymptomatic infection in a carrier situation, even in someone who is no longer showing any clinical signs.

‘virologically healed’ – this is a very unusual term? Virus may be undetectable in peripheral sites eg nasal and/or oropharyngeal swabs but there may still be residual virus replication/pathology at other (central) sites, hence the patient may have otherwise silent but virologically significant events taking place.

Can the authors express tissue culture data as TCID50 related to Cts graphically which would help the reader?

Another area the authors could expand on is application of sub-genomic RT-qPCR assays to unpick the molecular species that may signal in qPCR assays designed to detect viral RNA. Would the authors be able to shed any light in the species of RNA that is being detected eg full length, non-infectious RNA vs sub-genomic transcriptionally active RNA species. Other investigators have investigated these parameters (ie sub-genomic RNA, genomic RNA, culture). Ideally, all three are required to make a  secure call on viral infectivity status. This would greatly improve the manuscript.  

Contagiousness or infectiousness? Selection of one of these terms would be preferable eg infectiousness

Author Response

Response to Reviewer 3 Comments

Manzulli et al, compare real-time PCR detection of SARS-CoV-2 RNA and culture-based virus isolation to address the important question of viral infectivity during the acute and immediate post-acute infection period. Understanding the kinetics and dynamics of the SARS CoV-2 virus and hence implications for transmission remains an important question.
The data presented are derived from cohorts of SARS CoV-2 infected patients in Italy. The authors report on a ‘sensitive’ culture-based assay and compare their read-outs to a quantitative RT-PCR system. The data generated would appear to support the main finding of the paper although the authors do not present a completely convincing case and there appears to be no clear over-riding novel message driven by a unique data set. As the authors acknowledge other groups have reported that a PCR Ct value >24, in a patient >d8 post infection is likely to be non-infectious. As the outcome of any such study carries a significant public health message, complete rigour in scientific approach and execution must be applied.

Point 1: The methodology describing the culture side appeared to lack certain key details making it difficult for the reader to fully appraise the method used. The authors need to be more precise in their descriptions of methodology throughout.
Response 1: Thank you for the observation. As requested, we have added more details to the methodology that describe the culture side.
Point 2: ‘Clinically recovered’ is a fairly imprecise definition when dealing with potential asymptomatic infection in a carrier situation, even in someone who is no longer showing any clinical signs.
Response 2: Thank you for your observation. We agree with you, however according to the studies that have been published so far, it is quite improbable to become asymptomatic carrier after being symptomatic. We specify the definition in the text (lines 104-108).
Point 3: ‘virologically healed’ – this is a very unusual term? Virus may be undetectable in peripheral sites eg nasal and/or oropharyngeal swabs but there may still be residual virus replication/pathology at other (central) sites, hence the patient may have otherwise silent but virologically significant events taking place.
Response 3: Thank you for your interesting remark. We absolutely agree with this. In fact, what has been described so far, is that the persistence of viral replication in sites other than the respiratory system is not accompanied by actively infectious viral shedding, even in the intestinal mucosa, with active shedding through faeces, that might raise concern for inter-human transmission. This prevents the possibility to transmit the infection to other people with simple social contact. We corrected with “viral clearance from the respiratory system” (line 84).
Point 4: Can the authors express tissue culture data as TCID50 related to Cts graphically which would help the reader?
Response 4: Thanks for your question. We modified Table 1 trying to be more precise in the description between the cytopathic effect and the PCR value. It is our opinion that showing the exact Ct is a data not perfectly reproducible because it depends on the different sensitivity of the PCR kit used for the analyses.
Point 5: Another area the authors could expand on is application of sub-genomic RT-qPCR assays to unpick the molecular species that may signal in qPCR assays designed to detect viral RNA. Would the authors be able to shed any light in the
species of RNA that is being detected eg full length, non-infectious RNA vs sub-genomic transcriptionally active RNA species. Other investigators have investigated these parameters (ie sub-genomic RNA, genomic RNA, culture). Ideally, all three are required to make a secure call on viral infectivity status. This would greatly improve the manuscript.
Response 5: Thanks for the suggestion. We are going to perform another study based on WGS that will be able also to understand which kind of RNA we find. We think to submit a separate paper about this topic in the next future.
Point 6: Contagiousness or infectiousness? Selection of one of these terms would be preferable eg infectiousness
Response 6: Thank you for your observation. We used the term “infectiousness” in the text.

Round 2

Reviewer 2 Report

The authors have addressed comments and concerns and I consider the manuscript suitable for publication.

Author Response

Response to Reviewer 2 Comments

Point 1: The authors have addressed comments and concerns and I consider the manuscript suitable for publication.

Response 1: Thank you for your observations and considerations.

Reviewer 3 Report

While the authors have provided some additional information, unfortunately the manuscript is essentially unchanged in significant content.

Although the Table has been modified, it falls short of defining a precise relationship between CPE/culture data and PCR. If the authors cannot rely on Ct values, additional information should be provided to reflect differences in sensitivities of different PCR assays/kits that were used in the study. Did the authors include any run controls in their PCR assays that would assist in this interpretation? In the absence of fully validated PCR data, it makes comparison with culture very difficult to reliably interpret.  Due to issues of sensitivity, reproducibility etc, it therefore becomes especially important in a study of this nature to independently analyse all relevant biomarkers of infection. Hence, inclusion of sub-genomic RNA detection would in my opinion be a mandatory requirement to counterbalance the genomic RNA detection and culture data. I can only re-iterate that these data should be included in the analysis.

The authors suggest they are considering WGS (whole genome sequencing), however, this would not specifically address this question. As I am sure the authors are aware, deep sequence analysis of SARS CoV-2 provides information relating to the overall genetic identity of the infecting virus but cannot formally discern between infectious and non-infectious virus genomes (though may identify some that are defective) and, depending on the approach used, would likely have inherent issues with recovery of low copy genomes. However, these data if generated would be a useful adjunct and could be combined in a future article to provide a fuller account of the nature of the infecting viruses in the study population.  

Author Response

Response to Reviewer 3 Comments

While the authors have provided some additional information, unfortunately the manuscript is essentially unchanged in significant content.

Point 1: Although the Table has been modified, it falls short of defining a precise relationship between CPE/culture data and PCR. If the authors cannot rely on Ct values, additional information should be provided to reflect differences in sensitivities of different PCR assays/kits that were used in the study. Did the authors include any run controls in their PCR assays that would assist in this interpretation? In the absence of fully validated PCR data, it makes comparison with culture very difficult to reliably interpret.  Due to issues of sensitivity, reproducibility etc, it therefore becomes especially important in a study of this nature to independently analyse all relevant biomarkers of infection. Hence, inclusion of sub-genomic RNA detection would in my opinion be a mandatory requirement to counterbalance the genomic RNA detection and culture data. I can only re-iterate that these data should be included in the analysis.

Response 1: Thank you for your suggestion. We have added the table with the relationship between CPE/culture and PCR data. About sub-genomic analysis we agree with you that it would have provided useful information for the purposes of the study, but unfortunately it was not foreseen and could be a further aspect to be investigated in the future, although a recent study has highlighted some limits of this method. (Alexandersen S, Chamings A, Bhatta TR. SARS-CoV-2 genomic and subgenomic RNAs in diagnostic samples are not an indicator of active replication. Nat Commun. 2020 Nov 27;11(1):6059)

Point 2: The authors suggest they are considering WGS (whole genome sequencing), however, this would not specifically address this question. As I am sure the authors are aware, deep sequence analysis of SARS CoV-2 provides information relating to the overall genetic identity of the infecting virus but cannot formally discern between infectious and non-infectious virus genomes (though may identify some that are defective) and, depending on the approach used, would likely have inherent issues with recovery of low copy genomes. However, these data if generated would be a useful adjunct and could be combined in a future article to provide a fuller account of the nature of the infecting viruses in the study population. 

Response 2: Thank you for your observation. You’re right, but by WGS we would like to check the presence of whole genome viral particles or not and eventual modifications of the virus after the cultivation on cell lines.

We think that when we have cythopatic effect, the Ct values in PCR are much lower than the starting values of the swab. A late amplification in PCR cannot due to Sars-Cov-2 viral growth. On the contrary, a weak amplification could be associated to not infectious defective particles. In case of viral growth in cell culture, from our experience, we obtain very low Ct values (usually 10-15 Ct) and it’s impossible to obtain these values just for the presence of not infectious defective particles. At the end we always look at the presence of cythopatic effect on cells that can be caused just by the presence of infectious and virulent particles at high concentrations.